# Changes in Cortisol Secretion and Corticosteroid Receptors in COVID-19 and Non COVID-19 Critically Ill Patients with Sepsis/Septic Shock and Scope for Treatment

**DOI:** 10.3390/biomedicines11071801

**Published:** 2023-06-23

**Authors:** Ioannis Ilias, Alice G. Vassiliou, Chrysi Keskinidou, Charikleia S. Vrettou, Stylianos Orfanos, Anastasia Kotanidou, Ioanna Dimopoulou

**Affiliations:** 1Department of Endocrinology, Diabetes and Metabolism, Elena Venizelou Hospital, GR-11521 Athens, Greece; 21st Department of Critical Care Medicine and Pulmonary Services, School of Medicine, National and Kapodistrian University of Athens, Evangelismos Hospital, GR-10676 Athens, Greece; alvass75@gmail.com (A.G.V.); chrysakes29@gmail.com (C.K.); vrettou@hotmail.com (C.S.V.); stylianosorfanosuoa@gmail.com (S.O.); akotanid@med.uoa.gr (A.K.); idimo@otenet.gr (I.D.)

**Keywords:** CIRCI, sepsis, critical illness, adrenal, cortisol, glucocorticoid receptor

## Abstract

Sepsis is associated with dysregulated cortisol secretion, leading to abnormal levels of cortisol in the blood. In the early stages of the condition, cortisol levels are typically elevated due to increased secretion from the adrenal glands. However, as the disease progresses, cortisol levels may decline due to impaired adrenal function, leading to relative adrenal insufficiency. The latter is thought to be caused by a combination of factors, including impaired adrenal function, decreased production of corticotropin-releasing hormone (CRH) and adrenocorticotropic hormone (ACTH) by the hypothalamus and pituitary gland, and increased breakdown of cortisol. The dysregulation of cortisol secretion in sepsis is thought to contribute to the pathophysiology of the disease by impairing the body’s ability to mount an appropriate inflammatory response. Given the dysregulation of cortisol secretion and corticosteroid receptors in sepsis, there has been considerable interest in the use of steroids as a treatment. However, clinical trials have yielded mixed results and corticosteroid use in sepsis remains controversial. In this review, we will discuss the changes in cortisol secretion and corticosteroid receptors in critically ill patients with sepsis/septic shock. We will also make special note of COVID-19 patients, who presented a recent challenge for ICU management, and explore the scope for corticosteroid administration in both COVID-19 and non-COVID-19 septic patients.

## 1. Introduction

Critical illness refers to a state of poor health where the vital organs are not functioning properly and immediate care is necessary to prevent the risk of imminent death. This condition may however have the potential for reversal. While a broad spectrum of conditions can evolve to critical illness, sepsis and septic shock comprise the majority of cases and up to 30% of all ICU patients exhibit sepsis at some stage during their ICU stay [1]. The care of these patients involves a multidisciplinary approach and takes place in an intensive care unit (ICU) with experienced personnel [2]. Over time, critical illness and sepsis management has evolved from organ support and vital-sign monitoring to the identification of specific syndromes. Recently, biological heterogeneity within current critical states has been recognized through the findings of translational research [3].

Despite the fact that sepsis may have different etiologies, the pathophysiological pathways leading to septic shock and multiple organ failure are shared between different entities [4] and involve both immune and endocrine adaptive and maladaptive responses that evolve over time, in the acute, subacute, and chronic phase of patient care [5]. COVID-19-related critical illness displays many characteristics common in other septic syndromes with the predominance of respiratory system involvement, which may also include acute respiratory distress syndrome (ARDS) [6]. Table 1 summarizes the similarities and differences between COVID and non-COVID critical illness (Table 1).

The use of corticosteroids in critically ill patients has long been a matter of debate [17,18]. Even though the current evidence on their effects on outcomes remains inconclusive, they are broadly used in septic shock—particularly in cases with high dose vasopressors. The recent COVID-19 pandemic, and the widespread use of dexamethasone in these patients, revived the interest in corticosteroid effects during critical illness [19,20].

## 2. Critical Illness

### 2.1. Sepsis and Septic Shock

Sepsis is a critical condition characterized by dysfunction of the organs due to an uncontrolled response of the body to an infection [17]. Organ dysfunction can be evaluated with the sequential (sepsis-related) organ failure assessment (SOFA) score, and an increase in the SOFA score of two points or more is linked with an increase in mortality rate during hospitalization [21].

Septic shock is a specific condition characterized by severe circulatory, cellular, and metabolic dysfunction, leading to a higher risk of mortality compared to sepsis alone. The identification of septic shock in patients can be achieved through clinical evaluation, where a vasopressor is needed to maintain a mean arterial pressure of 65 mm Hg or higher and a serum lactate level greater than 2 mmol/L in the absence of hypovolemia. The combination of these two factors has been linked to hospital mortality rates exceeding 40%. Septic shock continues to be the primary cause of death in non-coronary ICUs [22].

### 2.2. HPA Axis and Critical Illness

Severe stress on the body is evident during critical illness. The stress response is primarily orchestrated by two major components, the sympathetic nervous system and the hypothalamic–pituitary–adrenal (HPA) axis [3]. Various stressor signals, such as hypoxia, hypotension, circulating angiotensin II, and cytokines, are conveyed to the amygdala [23]. The amygdala triggers a distress signal and the hypothalamus subsequently activates the sympathetic nervous system by transmitting signals through autonomic nerves to the adrenal glands. In response, these glands release epinephrine into the bloodstream, resulting in significant physiological changes that affect almost all organs. When stressors activate the HPA axis, the hypothalamic paraventricular nucleus releases corticotropin-releasing hormone (CRH), which, together with vasopressin, triggers the pituitary corticotropes to release adrenocorticotropic hormone (ACTH) into systemic circulation. This hormone acts on the adrenal gland’s fasciculate layer, which is responsible for the synthesis and secretion of glucocorticoids, with cortisol being the primary glucocorticoid in humans (Figure 1).

Cortisol binds to two receptor types, the glucocorticoid receptor (GCR) and the mineralocorticoid receptor (MR), to produce essential cardiovascular, metabolic, and immune regulatory responses necessary for survival. Glucocorticoids play a critical role in maintaining vascular tone, endothelial integrity, and vascular permeability during acute illness. Additionally, they enhance the vasoconstrictor effects of both endogenous and exogenous catecholamines.

### 2.3. Critical-Illness-Related Corticosteroid Insufficiency

Surviving critical illness depends on maintaining adequate adrenocortical function, which is typically reflected in elevated plasma cortisol levels among most critically ill patients. However, the increased systemic cortisol availability observed in critical illness is not solely attributable to a centrally activated HPA axis but rather to peripheral adaptations [24]. These adaptations include the release of circulating cortisol from plasma binding proteins, such as transcortin (CBG) and albumin, resulting in an increase in the free and active form of cortisol. Only the unbound “free” cortisol fraction is lipid soluble and can penetrate the cell membrane to bind with the cytoplasmic GCR. Additionally, cortisol breakdown in the liver and kidney is suppressed, further maintaining high levels of free cortisol in circulation and in the target tissues. In addition, an increase in CRH and ACTH, leading to high cortisol production, also plays a role [3].

Certain critically ill patients do not exhibit the expected increase in cortisol levels. This state was previously referred to as relative adrenal insufficiency (RAI), which means that cortisol production is inadequate in relation to the body’s needs [25].

The Surviving Sepsis Campaign guidelines from 2008 introduced the term critical-illness-related corticosteroid insufficiency (CIRCI) to describe a condition where the cellular activity of corticosteroids is inadequate for the severity of the patient’s illness [26]. CIRCI can be caused by a decrease in adrenal steroid production (adrenal insufficiency) or tissue resistance to glucocorticoids, leading to an exaggerated and prolonged proinflammatory response. The mechanisms responsible for HPA-axis dysfunction during critical illness are complex and may involve reduced production of CRH, ACTH, and cortisol, as well as dysfunction of their receptors. Damage to the adrenals caused by infarction or hemorrhage has also been suggested as a contributing factor. Although CIRCI may affect a broad range of critically ill patients, most research has focused on patients in the acute phase of septic shock. This condition may present clinically as hypotension and a lack of response to catecholamine infusions. The diagnosis of CIRCI is typically based on a delta total serum cortisol of less than 9 µg/dL after administration of 250 µg of i.v. cosyntropin or a random total cortisol of less than 10 µg/dL [27].

The 2017 update of the guidelines for the diagnosis and treatment of CIRCI by the Society of Critical Care Medicine (SCCM) and the European Society of Intensive Care Medicine (ESICM) acknowledged the complexity of the syndrome and questioned the reliability of a single diagnostic test [28]. The task force recommended the use of either random or delta cortisol for diagnosis, and suggested the high-dose (250-μg) ACTH stimulation test rather than the low-dose (1 μg) test. Although free cortisol is the active hormone, the authors proposed the use of plasma total cortisol due to its wider availability compared to free cortisol measurements. The use of salivary unbound cortisol was not recommended as it is not cost-effective or practical [28].

### 2.4. COVID-19 and Adrenal Function

Coronavirus disease 2019 (COVID-19), is caused by the severe acute respiratory syndrome coronavirus 2 (SARS-CoV-2). It predominantly affects the lungs but also other organs, including the endocrine glands [29]. The virus enters into cells through the angiotensin-converting enzyme 2 (ACE2) receptor, in the presence of transmembrane serine protease 2 (TMPRSS2) [29]. The endocrine system possesses both the requisite ACE2 receptor, and the TMPRSS2 protein necessary to permit the SARS-CoV-2 virion cellular access [29].

There are limited clinical data on HPA-axis function during acute COVID-19 infection, and these are derived from populations with varying disease severity. The major difficulty is that of performing a detailed evaluation of the HPA axis in patients with COVID-19 who are glucocorticoid dependent, i.e., those with respiratory failure or those treated within an ICU. Tan et al. first showed that patients with COVID-19 not receiving steroids mount a marked and appropriate acute cortisol stress response and that this response is significantly higher in COVID-19 compared to patients with clinical suspicion of COVID-19 that was not eventually confirmed [Tan et al.]. Furthermore, they found that elevated cortisol in COVID-19 cases was associated with an increased mortality and a reduced median survival, suggesting that cortisol probably reflects the severity of illness. Interestingly, cortisol seemed to be a better independent predictor than other laboratory markers associated with COVID-19, such as CRP, D-dimer, and the neutrophil-to-leukocyte ratio [30]. Tomo et al. recruited 76 RT-PCR-positive COVID-19 patients, including cases with severe lung involvement, and 79 healthy controls; the results showed increased levels of cortisol in COVID-19 cases compared with controls, as well as elevated cortisol levels in non-survivors [31]. In sharp contrast to the findings of the latter study, another research work found that patients with SARS-CoV-2 who had lower cortisol levels had a greater fatality rate. The authors applied a logistic regression model and found that a rise in cortisol levels by one unit correlated with a 26% lower mortality risk [32]. A meta-analysis showed that patients with severe COVID-19 had higher cortisol levels than patients with mild-to-moderate COVID-19; however, age and sex may affect this finding [33]. Overall, it seems that there is no consistent association between COVID-19 clinical outcomes and the presence of reduced or increased cortisol.

Clinical studies have suggested that adrenal insufficiency during COVID-19 may occur; according to a meta-analysis, its prevalence ranges from 3.1% to 64.3% in different studies [34]. Mechanisms leading to HPA-axis dysfunction are possibly multifactorial. Firstly, dysregulation of the HPA axis during the course of COVID-19 may be encountered as part of the development of functional CIRCI due to massive cytokine release. Indeed, in a small study, six of the nine COVID-19 critically ill patients had random plasma cortisol concentrations below 10 µg/dl, meeting the criteria for a diagnosis of CIRCI [35]. Iatrogenic causes resulting from prolonged treatment with synthetic glucocorticoids may also lead to HPA-axis dysfunction. Second, the expression of two receptors, ACE2 and TMPRSS2, have also been documented in the hypothalamus, pituitary, and adrenals, making them possible direct cytopathic targets of SARS-CoV-2. Adrenal small vessel necrosis and thrombosis, cortical lipid degeneration, endothellitis, and chronic inflammation have been described [36]. Similarly, in a postmortem study of COVID-19 patients, areas of pituitary necrosis/infarction have been reported [37]. Third, an interesting proposed mechanism is that antibodies produced by the host to counteract the virus may hinder the production of ACTH by the host, since there are similarities between certain amino acids of ACTH and those contained by the virus [38]. COVID-19 can lead to sepsis and septic shock. In one study, one out of three COVID-19 patients who were hospitalized had sepsis, which was mainly due to SARS-CoV-2; however, in 25% of cases, there was a concomitant bacterial infection [39]. COVID-19 patients with sepsis had a high mortality rate, particularly those with co-existing bacterial sepsis. These results confirm the significance of SARS-CoV-2 as a cause of sepsis and emphasize the importance of sepsis prevention and treatment in COVID-19. In an earlier meta-analysis, most patients with COVID-19 who required ICU admission exhibited infection-related organ dysfunction and met sepsis criteria [40]. These patients had a significantly higher mortality risk [41].

## 3. GCR Expression in Critical Illness and Sepsis, including COVID-19

### 3.1. The GCR

The actions of cortisol are mediated through two types of corticosteroid receptors: the MR and the GCR. The MR is primarily involved in regulating electrolyte balance, while the GCR plays a crucial role in regulating the immune response and inflammation. In sepsis, the expression and activity of both MR and GCR are altered, contributing to the pathophysiology of the disease. Studies have shown that MR expression is upregulated in sepsis, leading to increased sodium retention and fluid accumulation. This can lead to the development of edema, a common complication of sepsis. The GCR mediates the immunological, metabolic, and hemodynamic effects of endogenously produced and exogenously administered glucocorticoids. Alternative splicing of the primary transcript gives rise to two isoforms, GCR-α and GCR-β [42]. GCR-α is the receptor that binds to cortisol, whereas GCR-β has not been well described. Prior to cortisol-binding and translocation to the nucleus, the GCR-α resides in the cytoplasm in a large chaperone complex. The co-chaperone FK506 binding protein 5 (FKBP51), when bound to GCR-α, lowers its affinity for cortisol and negatively regulates the nuclear translocation of GCR-α [43]. After cortisol binding, GCR-α undergoes a conformational change leading to partial dissociation from the chaperone complex, exposing its two nuclear localization signals, and the complex moves from the cytosol to the nucleus to regulate gene transcription. This is achieved by binding to genes containing glucocorticoid-responsive elements (GREs) found in the promoters or intragenic regions of glucocorticoid-target genes [44]. Such genes include the glucocorticoid-inducible gene leucine zipper (GILZ) and serum/glucocorticoid regulated kinase 1 (SGK1), while negatively regulated genes include, amongst other inflammatory genes, β-arrestin, and osteocalcin (OSC) [45]. This transcriptional activation or repression ultimately results in the termination of the inflammatory response [46]. GCR-β has a C-terminal domain that cannot bind to natural or synthetic ligands and is known to suppress GCR-α activity [47,48,49]. Figure 2 diagrammatically represents cortisol signaling via GCR-α.

### 3.2. GCR Expression and Glucocorticoid Resistance in Critical Illness and Sepsis

The amount of circulating ligand and the tissue-specific expression of the enzyme that converts inert cortisone into metabolically active cortisol, namely 11β-hydroxysteroid dehydrogenase type 1 (11β-HSD1), define local glucocorticoid availability. The extent of the tissue-specific action of glucocorticoids and GCR depends on local cortisol cellular availability and the expression and function of the GCR. Glucocorticoid resistance refers to the inadequate response of the GCR to regulate the transcription of GCR-responsive genes, despite adequate plasma cortisol concentrations. Glucocorticoid resistance, and hence the magnitude of cortisol’s effect, may be due to decreased GCR-α mRNA and protein expression, the receptor subtype expressed, a reduced GCR affinity for cortisol and nuclear translocation, and/or binding to DNA [50]. Glucocorticoid resistance occurs in sepsis and may be a major contributor of the failure of glucocorticoids to improve septic patients. Evidence for an association between the degree of glucocorticoid unresponsiveness and disease severity and mortality has been demonstrated in acute respiratory distress response (ARDS) [51] and septic shock [52].

Most of the data on glucocorticoid resistance in critical illness are derived from experimental septic models. Endotoxin and lipopolysaccharide (LPS) injury models have shown a decreased ligand affinity and a down-regulation of GCR-α expression [53,54,55,56,57,58]. One group showed that impaired GCR-α dimerization resulted in worse lung barrier function during lipopolysaccharide (LPS)-induced inflammation and glucocorticoid treatment [59]. A study by the same group demonstrated that the impairment of GCR-α dimerization aggravated systemic hypotension and worsened lung function during LPS-induced endotoxic shock in mice [60]. Hence, they concluded that the GCR-α dimer is an important mediator of hemodynamic stability and lung function during LPS-induced systemic inflammation. Other animal models of sepsis have demonstrated the down-regulation of GCR-α and/or a decreased ligand affinity and up-regulation of GCR-β expression [58,61,62,63,64]. In a cecal ligation and puncture (CLP)-induced polymicrobial sepsis model, an intense initial activation of the GCR-α was noted prior to the induction of profound glucocorticoid resistance. The nuclear translocation of GCR and dexamethasone binding were not affected; however, DNA binding was affected. Hence, the authors suggested that the initial augmented GCR-α activity caused the unresponsiveness towards exogenously administered glucocorticoids seen later in the disease, since this initial GCR-α activation could “exhaust” the receptor [65]. Overall, it seems that GCR-α mRNA expression is downregulated, while GCR-β mRNA expression is up-regulated in the septic animal models.

Most human clinical studies have investigated cortisol availability in critical illness, with only a few exploring the role of GCR. The data from these studies suggest the existence of glucocorticoid resistance, especially in sepsis. More specifically, in septic patients, glucocorticoid treatment induced expression of miR-124, which in turn down-regulated GCR-α and limited the anti-inflammatory effects of glucocorticoids, prompting the authors to propose that steroid treatment might exacerbate GC resistance in patients with increased levels of GCR-β mRNA [66]. The expression of GCR-β in peripheral mononuclear cells of septic patients, and the effect of serum from septic patients on GCR expression and glucocorticoid sensitivity in cultured immune cells, has also been evaluated [67]. A transient increase in GCR-β mRNA expression was observed in sepsis, while septic patients’ sera induced glucocorticoid resistance in vitro [67]. Decreased GCR-α mRNA expression in peripheral blood mononuclear leucocytes in patients with sepsis or septic shock has been reported [68]. Another study examined GCR isoform abundance in tissues harvested from patients immediately after death from sepsis or non-septic critical illness, finding reduced GCR-α and elevated GCR-β receptor numbers in the heart and liver [61]. In septic shock, GCR-α expression was increased in T-lymphocytes, regardless of glucocorticoid treatment, while the GCR binding capacity was reduced in neutrophils of glucocorticoid-treated patients, suggesting a hampered response to exogenous or endogenous glucocorticoids since neutrophils are the predominant circulating leucocyte in septic shock [69]. More recently, septic non-survivors were shown to have a lower GCR-α expression and higher cortisol levels than septic survivors. Moreover, the septic patients exhibited upregulated plasma cortisol levels along with downregulated GCR-α expression in peripheral blood mononuclear cells (PBMCs) compared to controls, whereas GCR-β showed the opposite trend [70]. The reduced GCR-α expression and/or affinity in the blood of critically ill patients suggests that these patients develop glucocorticoid resistance, requiring higher doses of glucocorticoids. On the other hand, one study showed that, despite variation, the GCR number and affinity in mononuclear cells from patients during the hemodynamic compensatory phase of sepsis did not differ from control subjects, suggesting that glucocorticoids could be effective in the hemodynamic compensatory phase of sepsis [71]. Increased GCR-α mRNA and protein expression were shown in the acute phase of sepsis compared to systemic inflammatory response syndrome (SIRS) and healthy subjects, implying no requirement for exogenous steroid administration at this stage [72]. In one study on ventilated critically ill patients, decreased cytosolic GCR protein levels, and the subsequent downregulation of cortisol binding, were demonstrated before and after extubation [73]. Finally, our group showed that polymorphonuclear cells (PMNs) from critically ill patients who had not received steroids displayed a highly variable expression of GCR-α and GCR-β mRNA, with the expression levels of both receptors decreasing during ICU stays [74]. In the follow-up study, and compared to healthy controls, the mRNA expression of both GCR-α and GCR-β was increased, while during the sub-acute phase, the expression of both isoforms was lower compared to controls, as was the expression of FKBP5 and GILZ [75]. A recent report agrees with the results from these two studies. More specifically, Téblick and co-workers quantified the gene expression of key regulators of local glucocorticoid action, including 11β-HSD1, GCR-α, GCR-β, FKBP51, and GILZ, in various immune cells and tissues [76]. The expression profiles were assessed based on the duration of critical illness and glucocorticoid availability. In the patients’ neutrophils, GCR-α and GILZ were significantly suppressed during ICU stays, while low-to-normal GCR-α and increased GILZ expression were found in the patients’ monocytes. FKBP51 was increased in the monocytes and transiently increased in the neutrophils, whereas GCR-β was undetectable. In the septic patients, increased systemic glucocorticoid availability in most tissues was related to suppressed GCR-α, increased FKBP51, and unaltered GILZ. Only in the lung, and the adjacent diaphragm and adipose tissues, were increased circulating glucocorticoid levels found to lead to a higher GCR-α activity [76]. Thus, the authors proposed that the adaptations to critical illness that occur in specific tissues, independently of time, facilitate GCR-α action primarily to the lung, protecting against damage in other cells and tissues, including neutrophils. Throughout critical illness, GCR action was inhibited in neutrophils, possibly due to suppression of GCR-α expression; thus, glucocorticoid resistance could not be controlled by further increasing glucocorticoid availability. These findings do not seem to support the notion of a maladaptive generalized glucocorticoid resistance requiring treatment with glucocorticoids [76]. Finally, a novel human GCR variant, G459V, exhibited a hyperactive response when treated with hydrocortisone. More specifically, its activity was more than 30 times greater than the reference GCR-α. Unexpectedly, G459V showed considerably increased activity when treated with the GCR antagonist RU486. Thus, the authors concluded that variants of GCR can potentially alter the response to stress and steroid treatment, which could explain the mitigated clinical response in sepsis [77].

In respiratory syncytial virus (RSV) bronchiolitis-infected infants, the α:β GCR mRNA expression ratio was decreased significantly in severe disease compared to mild and normal controls. Furthermore, the expression of GCR-β positively correlated with the clinical score of severity, and might in part explain the insensitivity to glucocorticoid treatment in RSV infection [78]. Another study found significantly lower total and cytoplasmic, yet not nuclear, GCR-α protein levels in the PBMCs of critically ill children compared to healthy controls [79]. The authors suggested that the lower total and cytoplasmic receptor levels in critically ill children limit the GCR-mediated response to exogenous glucocorticoid therapy. The undisturbed translocation indicated that residual receptors retain their functionality and accessibility to therapeutic treatments. In another study in critically ill pediatric patients, those with shock and increased illness severity had lower GCR-α expression in CD4 and CD8 lymphocytes, while GCR-α expression did not correlate with cortisol levels [80]. Finally, in a different pediatric septic shock cohort, GCR-α expression did not differ between SIRS, sepsis, and septic shock. Decreased GCR-α protein expression, however, was found in septic shock patients with poor outcomes, especially those with increased cortisol levels [81].

### 3.3. GCR Εxpression in Severly and Critically Ill Patients with COVID-19

Data on COVID-19 and GCR are even more limited. Our group demonstrated that critically ill COVID-19 patients exhibited increased GCR-α and GILZ mRNA expression, and elevated cortisol levels, compared to equally severe non-COVID-19 critically ill patients [82]. Our results support the notion of the stimulation of the endogenous cortisol response to SARS-CoV-2, providing an additional rationale for corticotherapy in critically ill patients with COVID-19 that might, however, not be enough to prevent death [83]. Single-cell RNA sequencing data from the bronchoalveolar lavage fluid (BALF) of severe COVID-19 patients on corticosteroid treatment demonstrated that alveolar macrophages, smooth muscle cells, and endothelial cells co-express GCR and IL-6. GCR expression was decreased in severely ill COVID-19 patients compared to mild patients, prompting the authors to suggest that this may be a reflection of the pathological down-regulation of this endogenous immunomodulatory mechanism, which might be restored with corticosteroid therapy [84]. Very recently, it was demonstrated that in moderate-severe COVID-19 patients, GCR gene expression was significantly higher in those patients responding to corticosteroid treatment compared to the non-responders. GCR isoforms and mutations did not seem to correlate with the clinical response. Moreover, GILZ expression positively correlated with GCR expression. This study clarified the relationship between GCR expression and therapeutic responses to corticosteroids [85].

## 4. Current Evidence for Therapeutics: Clinical Trials and Future Research

Currently, the rationale for the use of corticosteroids in septic shock is based on the need to harness the exaggerated and impaired pro-inflammatory response during this condition. Corticosteroids are needed for normal cardiac function and have both inotropic and vascular actions. The inhibition of nitric oxide (NO) synthase by corticosteroids leads to the inhibition of sepsis-induced vasodilation mediated by NO [86].

Before the 1970s, high doses of glucocorticoids were empirically given to patients with septic shock. The perceived aim was to prevent an allergic or hypersensitivity reaction to the causative toxin, without the intention of replacing the adrenal cortex’s natural hormone production. At the time, the recommended dose was 300–500 mg of hydrocortisone every eight hours for three to five days [87].

The first organized trials investigating the use of glucocorticoids in septic shock were conducted in the 1970s and 1980s. One of the earliest randomized controlled trials (RCTs) investigating the use of glucocorticoids in septic shock was conducted in 1976. In this trial, septic shock patients were randomized either to receive 3 mg/kg dexamethasone or 30 mg/kg methylprednisolone (or the equivalent at multiple doses) or saline as placebo. A total of 254 patients with septic shock received dexamethasone (DEX) or methylprednisolone (MPS), and 246 received saline. In the latter, the mortality rate was 41%, whereas in the steroid-treated patients, mortality was significantly lower at 13%, with no difference between DEX vs. MPS [88]. Another early RCT was conducted by Sprung and colleagues in 1984. The study enrolled 59 patients with septic shock and randomized them to receive either DEX/MPS (n = 43) or placebo (n = 16). The study found a significant improvement in shock reversal with steroids but no improvement in survival versus placebo [89].

However, in the late 1980s and 1990s, it was generally agreed that corticosteroids should not be used to treat sepsis and septic shock due to studies showing no improvement in patient survival rates. Furthermore, the use of steroids in critically ill patients was feared, being linked to adverse effects, especially superinfections and polyneuromyopathy [90].

More recently, there has been an increased recognition of relative adrenal insufficiency in critically ill patients. The administration of glucocorticoids may counter the state of adrenal insufficiency induced by critical illness, which contributes to shock. Studies from the late 1990s and early 2000s have shown that lower doses of steroids administered over an extended period of time can provide hemodynamic benefits. Since then, numerous RCTs and meta-analyses have investigated the efficacy of glucocorticoids in septic shock, with varying results. The first major trial was the Corticosteroid Therapy of Septic Shock (CORTICUS) trial, which was conducted in 2008. This trial included 499 patients with septic shock (of whom 251 received hydrocortisone initially at 50 mg IV four times daily for 5 days and tapering for 6 days) and found no significant difference in mortality rates between those who received hydrocortisone and those who received a placebo [91,92,93,94]. Since the CORTICUS trial, several other trials have been conducted, including the Adjunctive Glucocorticoid Therapy in Patients with Septic Shock (ADRENAL) trial and the Activated Protein C and Corticosteroids for Human Septic Shock (APROCCHSS) trial [95,96,97,98].

In the ADRENAL trial, hydrocortisone was given at 200 mg/day for 7 days (or until discharge from the ICU); in the APROCCHSS trial, hydrocortisone 50 mg was intravenously administered four times daily (plus fludrocortisone 50 μg once daily) for 7 days. The ADRENAL trial included 3658 patients and found no significant difference in mortality rates between those who received hydrocortisone (n = 511/1832; 28%) and those who received a placebo (n = 526/1826; 29%). In contrast to these results, the APROCCHSS trial, which included 898 patients, found a significant reduction in mortality after 180 days in the trial with hydrocortisone treatment (47%) versus placebo (52%).

While these studies suggest that glucocorticoids may be beneficial—to a degree—in septic shock, their use remains controversial and future research is warranted. One area of research is the use of biomarkers to identify patients who are most likely to benefit from glucocorticoids.

Earlier studies found that patients with septic shock who had high levels of cortisol and low levels of ACTH at baseline were more likely to benefit from glucocorticoids [99]. Other research demonstrated that patients with septic shock who had higher levels of interleukin-6 (IL-6) at baseline were more likely to benefit from early goal-directed therapy [100]. These findings suggest that some biomarkers may be useful in identifying patients who are most likely to benefit from glucocorticoids.

Another area of research is the time of initiation, the dose, and the duration of glucocorticoid treatment. The CORTICUS, ADRENAL, and APROCCHSS studies used different hydrocortisone administration regimens. Future studies should further investigate the optimal dose and duration of glucocorticoid treatment in septic shock [86].

The concomitant administration of fludrocortisone with hydrocortisone is still being explored. A recent multicenter observational cohort study involving 88,275 patients with septic shock who were administered norepinephrine and initiated hydrocortisone treatment found that the inclusion of fludrocortisone alongside hydrocortisone was linked to a reduction of 3.7% in the difference between mortality or discharge rates compared to using hydrocortisone alone [101].

The role of CBG in regulating the action of glucocorticoids is essential. CBG regulates the availability of glucocorticoids by sequestering them in the bloodstream and affects the biological activity of glucocorticoids by modulating their binding to their intracellular receptors. The binding of glucocorticoids to CBG alters their conformation, which modulates their affinity for GCRs and, consequently, their ability to regulate gene expression [102]. By delving deeper into the role of CBG in inflammatory diseases in general, and in septic shock in particular, novel therapeutic approaches based on the concept of efficient delivery of cortisol to tissues could be discovered.

Gender vis-à-vis hydrocortisone administered in septic shock is another domain to be explored. The ADRENAL trial revealed that there were gender-based differences in the impact of hydrocortisone on sepsis. Specifically, hydrocortisone was found to increase the likelihood of shock recurrence in females, with an odds ratio of 1.48 [1.03–2.14; *p* = 0.03] [103].

The current state of medical knowledge and practice regarding the administration of DEX in ICU/critically ill patients with COVID-19 is that it can be an effective treatment to reduce mortality in certain patients. The use of DEX was first studied in the RECOVERY trial, a large randomized controlled trial in the UK that investigated several potential treatments for COVID-19 [104]. The trial found that DEX reduced mortality by about one-third in ventilated patients and by about one-fifth in patients receiving oxygen without mechanical ventilation. However, DEX was not found to be beneficial for patients who did not require respiratory support. Since the publication of the RECOVERY trial results, DEX has been recommended by several international guidelines for the treatment of COVID-19 in hospitalized patients, including those in the ICU. For example, the National Institutes of Health (NIH) COVID-19 Treatment Guidelines currently recommend the use of DEX in hospitalized patients who require supplemental oxygen or mechanical ventilation at a dose of 6 mg/day [12]. It is important to note that while DEX has shown promise in reducing mortality in certain patients, it is not a cure for COVID-19 and should be used judiciously.

## 5. Sepsis Guidelines

In 2008, the Surviving Sepsis Campaign guidelines recommended the use of hydrocortisone in patients with septic shock who were unresponsive to fluid resuscitation and vasopressor therapy. The guidelines suggested a dose of 200 mg/day of hydrocortisone for seven days. The use of other glucocorticoids, such as MPS and DEX, was not recommended due to the lack of evidence [26]. The 2021 Surviving Sepsis Campaign guidelines recommend the use of hydrocortisone in patients with septic shock and ongoing requirements for vasopressor therapy, but this recommendation is characterized as weak due to the moderate quality of the evidence. The typical dose used in adults is 200 mg/day given as 50 mg intravenously or as a continuous infusion. In addition, it is suggested for this to be commenced in cases requiring a dose of norepinephrine or epinephrine >0.25 mcg/Kg/min for at least 4 h [13].

## 6. Limitations

Our paper has certain limitations that we need to address. It is a narrative rather than a systematic review and reflects an approach to the relevant literature without analyzing the relevant evidence. Furthermore, the medical literature on cortisol response and GCR expression and function in critical illness and particularly in COVID disease is limited. Therefore, we cannot provide clear messages on the appropriateness of steroid use in these populations. Our work emphasizes the need for further research to identify the patients who would benefit from steroid supplementation on a physiological basis. Until then, clinicians are advised to follow existing published guidelines.

## 7. Conclusions

Sepsis and septic shock, also related to multiple organ failure, remain one of the main reasons for morbidity and mortality in critical illness. Adrenal pathology, and particularly CIRCI, has been described in this setting; nevertheless, the dysregulation of both steroid production and GCR expression and function contribute to hemodynamic compromise and organ dysfunction in both COVID and non-COVID patients. The administration of glucocorticoids in the management of sepsis and septic shock remains controversial, and current guidelines advise on their use in cases of severe hemodynamic compromise or documented adrenal insufficiency. While the use of corticosteroids may be beneficial in certain cases, their use should ideally be individualized, based on patient characteristics, as well as clinical criteria. Future research in this area should focus on identifying the patients who are most likely to benefit and determining the optimal dose and duration of treatment.

## Figures and Tables

**Figure 1 biomedicines-11-01801-f001:**
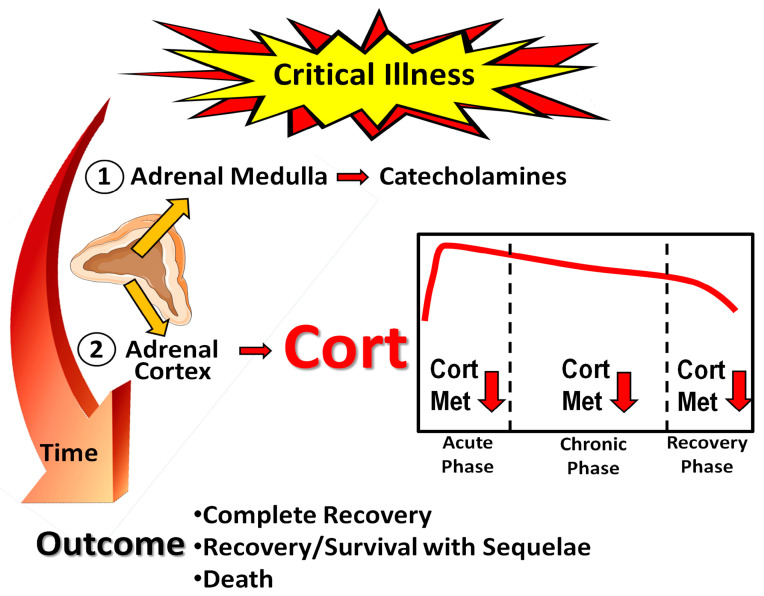
Schematic outline of the stress response to critical illness. Cort: cortisol; Cort Met: cortisol metabolism, including inactivation by 11β-Hydroxysteroid dehydrogenase type 2 to cortisone. Orange arrows and horizontal red arrows indicate hormone secretion and/or release, red arrows pointing downwards denote diminution. Parts of the figure were drawn using pictures from Servier Medical Art. Servier Medical Art by Servier is licensed under a Creative Commons Attribution 3.0 Unported License (https://creativecommons.org/licenses/by/3.0/, accessed on 15 May 2023).

**Figure 2 biomedicines-11-01801-f002:**
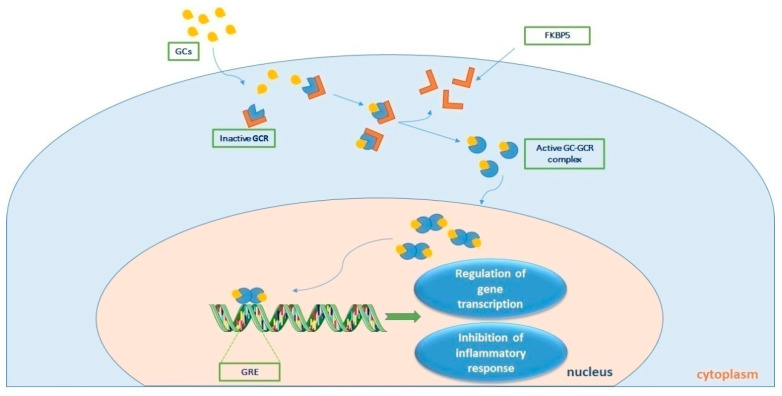
Τhe process of cortisol signaling via the glucocorticoid receptor (GCR). Upon binding with cortisol, the GCR-cortisol complex is transported from the cytosol to the nucleus. Once in the nucleus, the complex influences transcriptional activation or repression by directly binding to genes containing glucocorticoid (GC) responsive elements (GREs), ultimately leading to the suppression of the inflammatory response. GC-GCR: cortisol-glucocorticoid receptor complex; FKBP5 is a co-chaperone of the GCR. Parts of the figure were drawn by using pictures from Servier Medical Art. Servier Medical Art by Servier is licensed under a Creative Commons Attribution 3.0 Unported License (https://creativecommons.org/licenses/by/3.0/, accessed on 15 May 2023).

**Table 1 biomedicines-11-01801-t001:** Comparison of characteristics between COVID-19-related sepsis and sepsis of different etiologies.

Clinical and Laboratory Characteristics	COVID-19	Other Etiologies
Cultured pathogens [7]	Initially (−)	Initially (+) in most cases
Cytokine storm [8,9]	+/−	+/−
T-cell deficits [8,9]	+/−	+/−
Immunosupression profile [8,9]	+/−	+/−
TNFα/IL-1b [10,11]	↑↑	↑
Interferon responses [10,11]	↓	↔
Plasma cortisol	↑, ↓, or ↔	↑, ↓, or ↔
CIRCI	+/−	+/−
GCR	GCR-α ↑ or ↓	GCR-α mostly ↓, ↓ ligand affinity for GCR-β
Steroid treatment	DEX * [12]	HC * [13]
Long term outcomes	Long COVID syndrome [14]	Post-sepsis syndrome [15]
Mortality [16]	Higher compared to non-COVID	33–52%

↑: increase, ↑↑: increase to a large degree, ↔: no change, ↓: decrease, (−): negative, (+): positive, +: noted, −: not noted, DEX: dexamethasone, HC: hydrocortisone, * according to guidelines and for selected patients.

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
