# Peer review of "Changes in Cortisol Secretion and Corticosteroid Receptors in COVID-19 and Non COVID-19 Critically Ill Patients with Sepsis/Septic Shock and Scope for Treatment"

_biomedicines, 2023, doi:10.3390/biomedicines11071801_

Round 1

Reviewer 1 Report

The following changes must be included in the manuscript “Changes in cortisol secretion and corticosteroid receptors in critically ill patients with sepsis/septic shock and scope for treatment” to consider it for acceptance.

  1. It is suggested to modify the abstract to include details about the paper and what it is intended for. The abstract must specify what the authors intend to discuss in  the manuscript.

  2. The introduction seems vague. The introduction should clearly state the purpose of the manuscript and introduce the readers towards upcoming sections discussed in the review. It is suggested to rewrite it.

  3. The flow of the write up needs to be modified. The sudden inclusion of COVID-19 does not serve what the title of the manuscript is intended to deliver. So, change the title and correct the flow of the article. Also, paragraphs 2.5 and 2.4 should be merged into one under the single title “COVID-19 and Sepsis/Septic Shock” followed by 3.3 and then point 6. 

  4. The conclusion is also not at par. It should be modified to critically discuss the outcome of the review and what purpose it serves.

Author Response

Reviewer 1
We thank the Reviewer for the insightful comments and suggestions. Our manuscript was modified accordingly - please see below.

[1]. It is suggested to modify the abstract to include details about the paper and what it is intended for. The abstract must specify what the authors intend to discuss in  the manuscript.

The abstract has been changed in the revised version of the manucript as follows:
"Sepsis is associated with dysregulated cortisol secretion, leading to abnormal levels of cortisol in the blood. In the early stages of the condition, cortisol levels are typically elevated due to increased secretion from the adrenal glands. However, as the disease progresses, cortisol levels may decline due to impaired adrenal function, leading to relative adrenal insufficiency. The latter is thought to be caused by a combination of factors, including impaired adrenal function, decreased production of corticotropin-releasing hormone (CRH) and adrenocorticotropic hormone (ACTH) by the hypothalamus and pituitary gland, and increased breakdown of cortisol. The dysregulation of cortisol secretion in sepsis is thought to contribute to the pathophysiology of the disease by impairing the body's ability to mount an appropriate inflammatory response. Given the dysregulation of cortisol secretion and corticosteroid receptors in sepsis, there has been considerable interest in the use of steroids as a treatment. However, clinical trials have yielded mixed results, and corticosteroid use in sepsis, remains controversial. 
In this review we will discuss the changes in cortisol secretion and corticosteroid receptors in critically ill patients with sepsis/septic shock. We will also make special note of COVID-19 patients, who presented a recent challenge for ICU management, and explore the scope for corticosteroid administration in both COVID-19 and non-COVID-19 septic patients."

[2]. The introduction seems vague. The introduction should clearly state the purpose of the manuscript and introduce the readers towards upcoming sections discussed in the review. It is suggested to rewrite it.

The introduction has been changed in the revised version of the manuscript as follows:
"Critical illness refers to a state of poor health where the vital organs are not func-tioning properly, and immediate care is necessary to prevent the risk of imminent death. This condition may however have the potential for reversal.. While a broad spectrum of conditions can evolve to critical illness, sepsis and septic shock comprise the majority of cases, and up to 30% of all ICU patients have sepsis at some stage during their ICU stay [1]. The care of these patients involves a multidisciplinary approach and takes place in an intensive care unit (ICU) with experienced personnel. [2]. Over time, critical illness and sepsis management has evolved from organ support and vital sign monitoring to the identification of specific syndromes. Recently, biological heterogeneity within current critical states has been recognized through findings of translational research [3].
 Despite the fact that sepsis may have different etiologies, the pathophysiological pathways leading to septic shock and multiple organ failure are shared between different entities [4] and involve both immune and endocrine adaptive and maladaptive responses that evolve over time, in the acute, subacute and chronic phase of patient care [5]. COVID-19 related critical illness has many characteristics common with other septic syn-dromes, with the predominance of respiratory system involvement, that may also include Acute Respiratory Distress Syndrome (ARDS) [6]. Table 1 summarizes the similarities and differences between COVID and non- COVID critical illness (Table 1).
    The use of corticosteroids in critically ill patients has been a long matter of debate [17,18]. Even though current evidence on their effects on outcomes remains inconclusive, they are broadly used in septic shock, particularly in cases with high dose vasopressors. The recent COVID-19 pandemic, and the widespread use of dexamethasone in these pa-tients, revived the interest in corticosteroid effects during critical illness [19,20]."

[3]. The flow of the write up needs to be modified. The sudden inclusion of COVID-19 does not serve what the title of the manuscript is intended to deliver. So, change the title and correct the flow of the article. Also, paragraphs 2.5 and 2.4 should be merged into one under the single title “COVID-19 and Sepsis/Septic Shock” followed by 3.3 and then point 6. 

These issues were addressed in the revised version of the article. The title has been changed to "Changes in cortisol secretion and corticosteroid receptors in COVID and non COVID critically ill patients with sepsis/septic shock and scope for treatment" while the introduction has been extensively changed, a new table has been added and the new section 2.4 is as follows:
"2.4.COVID-19 and Adrenal function
The coronavirus disease 2019 (COVID?19), is caused by the severe acute respiratory syndrome coronavirus 2 (SARS?CoV?2). It affects predominantly the lungs but also other organs, including the endocrine glands [29]. The virus enters into the cells through the angiotensin-converting enzyme 2 (ACE2) receptor, in the presence of transmembrane se-rine protease 2 (TMPRSS2) [29]. The endocrine system possesses both the requisite ACE2 receptor, and the TMPRSS2 protein necessary to permit the SARS-CoV-2 virion cellular access [29].
There are limited clinical data on HPA axis function during acute COVID-19 infec-tion, and they are derived from populations with varying disease severity . The major difficulty is that of performing a detailed evaluation of the HPA axis in patients with COVID-19 who are glucocorticoid dependent, i.e. those with respiratory failure or those treated within an ICU. Tan et al first showed that patients with COVID-19 not receiving steroids mount a marked and appropriate acute cortisol stress response and that this response is significantly higher in COVID-19 compared to patients with clinical suspicion of COVID-19 that was not eventually confirmed [Tan et al]. Furthermore, they found that elevated cortisol in COVID-19 cases, was associated with increased mortality and a re-duced median survival, suggesting that cortisol probably reflects severity of illness. In-terestingly, , cortisol seemed to be a better independent predictor than were other labor-atory markers associated with COVID-19, such as CRP, D-dimer, and neutrophil to leu-kocyte ratio [30]. Tomo et al recruited 76 RT-PCR-positive COVID-19 patients, including cases with severe lung involvement, and 79 healthy controls and showed increased levels of cortisol in COVID-19 cases compared with controls, as well as elevated cortisol levels in non-survivors [31]. In sharp contrast to the findings of the latter study, another research work found that patients with SARS-CoV-2 who had lower cortisol levels had a greater fatality rate. The authors applied a logistic regression model and found that a rise in cortisol levels by one unit correlated with a 26% lower mortality risk [32]. A meta-analysis showed that patients with severe COVID-19 had higher cortisol levels than patients with mild-to-moderate COVID-19, however, age and sex may affect this finding [33]. Overall, it seems that there is no consistent association between COVID-19 clinical outcome and the presence of reduced or increased cortisol. 
Clinical studies suggest that adrenal insufficiency during COVID-19 may occur, and according to a meta-analysis its prevalence ranges from 3.1% to 64.3% in different studies [34]. Mechanisms leading to HPA axis dysfunction are possibly multifactorial. Firstly, dysregulation of the HPA axis during the course of COVID-19 may be encountered as part of the development of functional CIRCI due to massive cytokines release. Indeed, in a small study, six of the nine COVID-19 critically ill patients had random plasma cortisol concentrations below 10 µg/dl, meeting the criteria for the diagnosis of CIRCI [35]. Ia-trogenic causes resulting from prolonged treatment with synthetic glucocorticoids may also lead to HPA axis dysfunction. Second, the expression of two receptors, ACE2 and TMPRSS2, have also been documented in the hypothalamus, pituitary and the adrenals, making them possible direct cytopathic targets of SARS-CoV-2. Adrenal small vessel ne-crosis, and thrombosis, cortical lipid degeneration, endothellitis, and chronic inflamma-tion have been described [36]. Similarly, in a postmortem study of COVID-19 patients, areas of pituitary necrosis/infarction have been reported [37]. Third, an interesting pro-posed mechanism, is that antibodies produced by the host to counteract the virus, may hinder the production of ACTH by the host, since there are similarities between certain amino acids of ACTH and those contained by the virus  [38]COVID-19 can lead to sepsis and septic shock. In one study, one out of three COVID-19 patients who were hospitalized had sepsis, which was mainly due to SARS-CoV-2, but in 25% of the cases, there was concomitant bacterial infection [39]. COVID-19 patients with sepsis had a high mortality rate, particularly those with co-existing bacterial sepsis. These results confirm the significance of SARS-CoV-2 as a cause of sepsis and emphasize the importance of sepsis prevention and treatment in COVID-19. In an earlier metaanalysis, most patients with COVID-19 who required ICU admission exhibited infection-related organ dysfunction and met sepsis criteria [40]. These patients had significantly higher mortality risk. [41]." 

[4]. The conclusion is also not at par. It should be modified to critically discuss the outcome of the review and what purpose it serves.

The revised conclusion is as follows:
"7. Conclusion
Sepsis and septic shock, also related to multiple organ failure, remains one of the main reasons for morbidity and mortality in critical illness. Adrenal pathology, and par-ticularly CIRCI has been described in this setting, nevertheless, dysregulation of both steroid production and GCR expression and function, are contributing to hemodynamic compromise and organ dysfunction, in both COVID and non-COVID patients. The ad-ministration of glucocorticoids in the management of sepsis and septic shock, remains controversial, and current guidelines advise on their use in cases of severe hemodynamic compromise or documented adrenal insufficiency. While the use of corticosteroids may be beneficial in some cases, their use should ideally be individualized, based on patient characteristics as well as clinical criteria. Future research in this area should focus on identifying the patients who are most likely to benefit, and determining the optimal dose and duration of treatment."

Reviewer 2 Report

The review focused on cortisol secretion and cortisol receptors in critically ill patients. Although the subject is enjoyable, some issues should be addressed. 

For instance, there is no indication of COVID-19 in the title.

Sepsis has a different etiology, and the initial infectious focus may vary, implication diverse clinical outcomes. It was not discussed in the text. Sepsis can originate from viral, bacterial, fungal, and parasite infections. Also, sepsis may evolve to a hipper inflammatory state or an immunosuppressive state, and it was not discussed. A timeline of disease is critical to understand how to manage the patient. There is no timeline in the review. For me, it is mandatory to indicate it. 

The second major issue is the introduction of COVID-19 in 2.4. It should be headed as sepsis and adrenal function. At this point, it is confusing COVID-19 appeared here because severe COVID-19 is diagnosed as sepsis, but with some differences. If the authors intend to compare general (bacterial, viral) sepsis to COVID-19, it should be better introduced. Also, as sepsis COVID-19 pathogenesis is rather complex, and typical COVID-19 patient evolves poor outcome does not have the virus, and they die from severe organ dysfunction or due to secondary infection. In the case of the authors intend to compare sepsis and COVID-19, some significant modifications are required. The authors should consider including a table comparing sepsis x COVID clinical evolution during the time, similarities, and differences between them. In addition, indicate in each study regarding cortisol and CGR the degree of severity of COVID-19 patients because COVID-19 can have a vast spectrum of clinical conditions.

Minor

Change F to C in Figure 1. 

Include a summary figure with the main findings. 

Also, include a study limitation section. 

The English could benefit from a revision by a native speaker. 

Author Response

Reviewer 2
We thank the Reviewer for the insightful comments and suggestions. Our manuscript was modified accordingly - please see below.

[1]. The review focused on cortisol secretion and cortisol receptors in critically ill patients. Although the subject is enjoyable, some issues should be addressed. For instance, there is no indication of COVID-19 in the title.

The title has been changed to "Changes in cortisol secretion and corticosteroid receptors in COVID and non COVID critically ill patients with sepsis/septic shock and scope for treatment" 

[2]. Sepsis has a different etiology, and the initial infectious focus may vary, implication diverse clinical outcomes. It was not discussed in the text. Sepsis can originate from viral, bacterial, fungal, and parasite infections. Also, sepsis may evolve to a hipper inflammatory state or an immunosuppressive state, and it was not discussed. A timeline of disease is critical to understand how to manage the patient. There is no timeline in the review. For me, it is mandatory to indicate it. 

These points have been addressed in the revised introduction section and the new section 2.4 as follows:
"1. Introduction
Critical illness refers to a state of poor health where the vital organs are not func-tioning properly, and immediate care is necessary to prevent the risk of imminent death. This condition may however have the potential for reversal.. While a broad spectrum of conditions can evolve to critical illness, sepsis and septic shock comprise the majority of cases, and up to 30% of all ICU patients have sepsis at some stage during their ICU stay [1]. The care of these patients involves a multidisciplinary approach and takes place in an intensive care unit (ICU) with experienced personnel. [2]. Over time, critical illness and sepsis management has evolved from organ support and vital sign monitoring to the identification of specific syndromes. Recently, biological heterogeneity within current critical states has been recognized through findings of translational research [3].
 Despite the fact that sepsis may have different etiologies, the pathophysiological pathways leading to septic shock and multiple organ failure are shared between different entities [4] and involve both immune and endocrine adaptive and maladaptive responses that evolve over time, in the acute, subacute and chronic phase of patient care [5]. COVID-19 related critical illness has many characteristics common with other septic syn-dromes, with the predominance of respiratory system involvement, that may also include Acute Respiratory Distress Syndrome (ARDS) [6]. Table 1 summarizes the similarities and differences between COVID and non- COVID critical illness (Table 1).
    The use of corticosteroids in critically ill patients has been a long matter of debate [17,18]. Even though current evidence on their effects on outcomes remains inconclusive, they are broadly used in septic shock, particularly in cases with high dose vasopressors. The recent COVID-19 pandemic, and the widespread use of dexamethasone in these pa-tients, revived the interest in corticosteroid effects during critical illness [19,20]."

&

"2.4.COVID-19 and Adrenal function
The coronavirus disease 2019 (COVID?19), is caused by the severe acute respiratory syndrome coronavirus 2 (SARS?CoV?2). It affects predominantly the lungs but also other organs, including the endocrine glands [29]. The virus enters into the cells through the angiotensin-converting enzyme 2 (ACE2) receptor, in the presence of transmembrane se-rine protease 2 (TMPRSS2) [29]. The endocrine system possesses both the requisite ACE2 receptor, and the TMPRSS2 protein necessary to permit the SARS-CoV-2 virion cellular access [29].
There are limited clinical data on HPA axis function during acute COVID-19 infec-tion, and they are derived from populations with varying disease severity . The major difficulty is that of performing a detailed evaluation of the HPA axis in patients with COVID-19 who are glucocorticoid dependent, i.e. those with respiratory failure or those treated within an ICU. Tan et al first showed that patients with COVID-19 not receiving steroids mount a marked and appropriate acute cortisol stress response and that this response is significantly higher in COVID-19 compared to patients with clinical suspicion of COVID-19 that was not eventually confirmed [Tan et al]. Furthermore, they found that elevated cortisol in COVID-19 cases, was associated with increased mortality and a re-duced median survival, suggesting that cortisol probably reflects severity of illness. In-terestingly, , cortisol seemed to be a better independent predictor than were other labor-atory markers associated with COVID-19, such as CRP, D-dimer, and neutrophil to leu-kocyte ratio [30]. Tomo et al recruited 76 RT-PCR-positive COVID-19 patients, including cases with severe lung involvement, and 79 healthy controls and showed increased levels of cortisol in COVID-19 cases compared with controls, as well as elevated cortisol levels in non-survivors [31]. In sharp contrast to the findings of the latter study, another research work found that patients with SARS-CoV-2 who had lower cortisol levels had a greater fatality rate. The authors applied a logistic regression model and found that a rise in cortisol levels by one unit correlated with a 26% lower mortality risk [32]. A meta-analysis showed that patients with severe COVID-19 had higher cortisol levels than patients with mild-to-moderate COVID-19, however, age and sex may affect this finding [33]. Overall, it seems that there is no consistent association between COVID-19 clinical outcome and the presence of reduced or increased cortisol. 
Clinical studies suggest that adrenal insufficiency during COVID-19 may occur, and according to a meta-analysis its prevalence ranges from 3.1% to 64.3% in different studies [34]. Mechanisms leading to HPA axis dysfunction are possibly multifactorial. Firstly, dysregulation of the HPA axis during the course of COVID-19 may be encountered as part of the development of functional CIRCI due to massive cytokines release. Indeed, in a small study, six of the nine COVID-19 critically ill patients had random plasma cortisol concentrations below 10 µg/dl, meeting the criteria for the diagnosis of CIRCI [35]. Ia-trogenic causes resulting from prolonged treatment with synthetic glucocorticoids may also lead to HPA axis dysfunction. Second, the expression of two receptors, ACE2 and TMPRSS2, have also been documented in the hypothalamus, pituitary and the adrenals, making them possible direct cytopathic targets of SARS-CoV-2. Adrenal small vessel ne-crosis, and thrombosis, cortical lipid degeneration, endothellitis, and chronic inflamma-tion have been described [36]. Similarly, in a postmortem study of COVID-19 patients, areas of pituitary necrosis/infarction have been reported [37]. Third, an interesting pro-posed mechanism, is that antibodies produced by the host to counteract the virus, may hinder the production of ACTH by the host, since there are similarities between certain amino acids of ACTH and those contained by the virus  [38]COVID-19 can lead to sepsis and septic shock. In one study, one out of three COVID-19 patients who were hospitalized had sepsis, which was mainly due to SARS-CoV-2, but in 25% of the cases, there was concomitant bacterial infection [39]. COVID-19 patients with sepsis had a high mortality rate, particularly those with co-existing bacterial sepsis. These results confirm the significance of SARS-CoV-2 as a cause of sepsis and emphasize the importance of sepsis prevention and treatment in COVID-19. In an earlier metaanalysis, most patients with COVID-19 who required ICU admission exhibited infection-related organ dysfunction and met sepsis criteria [40]. These patients had significantly higher mortality risk. [41]." 

[3]. The second major issue is the introduction of COVID-19 in 2.4. It should be headed as sepsis and adrenal function. At this point, it is confusing COVID-19 appeared here because severe COVID-19 is diagnosed as sepsis, but with some differences. If the authors intend to compare general (bacterial, viral) sepsis to COVID-19, it should be better introduced. Also, as sepsis COVID-19 pathogenesis is rather complex, and typical COVID-19 patient evolves poor outcome does not have the virus, and they die from severe organ dysfunction or due to secondary infection. In the case of the authors intend to compare sepsis and COVID-19, some significant modifications are required. The authors should consider including a table comparing sepsis x COVID clinical evolution during the time, similarities, and differences between them. In addition, indicate in each study regarding cortisol and CGR the degree of severity of COVID-19 patients because COVID-19 can have a vast spectrum of clinical conditions.

This issue has been addressed in the new section 2.4 [please see above response to point 2]. Additionally please also see the new section 3.3 as follows:
"3.3. GCR Εxpression in severly and critically ill patients with COVID-19
Data on COVID-19 and GCR are even more limited. Our group demonstrated that critically ill COVID-19 patients exhibited increased GCR-α and GILZ mRNA expression, and elevated cortisol levels, compared to equally severe non-COVID-19 critically ill pa-tients [82]. Our results support the notion of the stimulation of the endogenous cortisol response to SARS-CoV-2, providing additional rationale for corticotherapy in critically ill patients with COVID-19 that might, however, not be enough to prevent death [83]. Sin-gle-cell RNA sequencing data from bronchoalveolar lavage fluid (BALF) of severe COVID-19 patients on corticosteroid treatment demonstrated that alveolar macrophages, smooth muscle cells, and endothelial cells co-express GCR and IL-6. GCR expression was decreased in severely ill COVID-19 patients compared to mild patients, prompting the authors to suggest that this may be a reflection of the pathological down-regulation of this endogenous immunomodulatory mechanism, which might be restored with corti-costeroid therapy [84]. Very recently it was demonstrated that in moderate-severe COVID-19 patients, GCR gene expression was significantly higher in the patients re-sponding to corticosteroid treatment compared to the non-responders. GCR isoforms and mutations did not seem to correlate with clinical response. Moreover, GILZ expression positively correlated with GCR expression. This study clarified the relationship between GCR expression with therapeutic responses to corticosteroids [85]." ---- please note that the cited references in this section pertain to critical illness, with the exception of Ref [85], which also included moderately ill patients.

Minor
[4]. Change F to C in Figure 1. 

We had initially chosen the classical abbreviation "F" for cortisol [as introduced by Kendall & Reichstein and advocated by H Raff in Endocrinology 2016; 157: 3307-3308] - in the revised version of the figure we changed "F" to "Cort".  

[5]. Include a summary figure with the main findings. 

In the revised version of the manuscript we have added a table that summarizes the main findings. 

[6]. Also, include a study limitation section. 

In the revised version of the manuscript we have added a limitations section as follows:
"6. Limitations.
Our paper has some limitations that we need to address. It ia a narrative rather than a systematic review, and reflects an approach to the relevant literature without analysisng the relevant evidence. Furthermore, the medical literature on cortisol response and GCR expression and function in critical illness and particularly in COVID disease is limited. Therefore, we cannot come up with clear messages on the appropriateness of steroid use in this populations. Our work emphasizes the need for further research to identify the patients who would benefit from steroid supplementations on a physiological basis. Until then, clinicians are advised to follow existing published guidelines." 

Comments on the Quality of English Language
[7]. The English could benefit from a revision by a native speaker. 

The text has been re-checked by Dr A.G. Vassiliou, who is a native English speaker from Australia.

Round 2

Reviewer 1 Report

The modified manuscript seems fine. The suggested changes have been done.

Reviewer 2 Report

The authors have addressed the raised issues. 

The authors made some revisions.